# Research Advances of Soil Corrosion of Grounding Grids

**DOI:** 10.3390/mi12050513

**Published:** 2021-05-02

**Authors:** Cheng Zhang, Yuxiang Liao, Xue Gao, Jing Zhao, Yuan Yuan, Ruijin Liao

**Affiliations:** 1College of Materials Science and Engineering, Chongqing University, Chongqing 400044, China; 20163014@cqu.edu.cn (C.Z.); gaoxue@cqu.edu.cn (X.G.); 2State Grid Chongqing Electric Power Research Institute, Chongqing 401123, China; liaoyuxiang@cq.sgcc.com.cn (Y.L.); zhaojing_sgcc@163.com (J.Z.); 3The State Key Laboratory of Power Transmission Equipment & System Security and New Technology, Chongqing University, Chongqing 400044, China; rjliao@cqu.edu.cn

**Keywords:** grounding grid, soil corrosion, stray current corrosion, detection technology, protective measures

## Abstract

A grounding grid plays the role of discharging current and balancing voltage to ensure the safety of the power system. However, soil corrosion can damage the grounding grid, which then can endanger the safe operation of power system. This paper reviewed recent research advances of soil corrosion of grounding grid. The cause, mechanism, types, and influencing factors of soil corrosion of grounding grids were summarized, and the corresponding detection technology and protective measures were also introduced. The paper pointed out that soil corrosion is a serious threat to the grounding grid system. Moreover, the impact mechanism of AC stray current, new corrosion detection technology, and better protective measures still need in-depth research.

## 1. Introduction

The grounding grid is an important device that ensures the safe operation of the power system; it plays the role of balancing voltage and discharging current in the event of a lightning strike or equipment failure [1,2]. However, due to the grounding conductor being buried underground for a long term, the corrosion caused by the soil environment and the working conditions inevitably can lead to the destruction of the grounding conductor [3,4,5]. The corrosion of the grounding conductor is shown in Figure 1. When the ground conductor fault occurs in the power system, the current cannot fully spread in the soil. In this case, the ground resistance and ground potential rise, and high voltage is connected to the secondary circuit, which affects the safe operation of electrical equipment and personal safety, causing major accidents and economic losses [3,6,7].

There has been research on the corrosion of grounding grids for decades. Back in the 1950s, scholars began to pay attention to the detection and protection of grounding grid corrosion. Rajan et al. [8] analyzed the relationship of corrosion and grounding. From there, basic principles and necessary precautions were given to the grounding grid designer. Lawson et al. [9] proposed four tests to estimate the rates of a substation grounding grid and guy anchor corrosion. The degree of corrosion and the form of correction necessary were given. However, grounding grid corrosion is still a worldwide safety hazard in the power system until recent years. According to the data analysis of the worldwide investigation agency, the annual corrosion rate of a grounding grid conductor in corrosive soil can reach 2.0 mm; in the more corrosive soil, the annual corrosion rate can reach 3.4 mm, and in the extremely corrosive soil, its annual corrosion rate can even reach 8.0 mm. On May 25, 2005, due to the aging and partial corrosion of the transformer grounding line in the southern part of Moscow, Russia, all 321 substations of the Moscow power grid were shut down, and as a result, the industrial production, commercial activities, and transportation in about half of Moscow city were paralyzed [10]. In 2008, the neutral point potential of a 330-kV substation in Northern Xi’an, China, increased abnormally, and the DC system was destroyed, causing a large-scale blackout [11]. After inspection, it was found that the cause of the accident was the internal corrosion of the grounding grid, and the neutral point could not be grounded normally. On November 11, 2009, due to a grounding grid failure caused by a lightning strike, a large-scale blackout occurred across the Brazilian power grid, affecting approximately 50 million people [12]. The loss of load caused by this accident is about 24,436 MW, which accounts for roughly 40% of the total load of Brazil′s power grid. In September 2011, major power outages occurred in the United States and Mexico. The cause of the accident was the serious corrosion of the grounding device, the aging of the connection line, and the serious load of the power grid [13]. In 2017, Guangdong, China, after excavation and detection, it was discovered that the conductor of the Huiyang 220 kV voltage level substation, which had been in operation for 26 years, was corroded by 70% and locally broken [14]. The hazard and the universality of grounding grid corrosion are well illustrated by the above accident examples.

With the rapid development of the power system, the scale and structure of power grids have improved significantly compared with those in the initial period. The power system has developed in the direction of ultrahigh voltage, ultralong-distance transportation, and ultralarge capacity. Therefore, higher requirements on the operational performance of grounding grids were put forward. However, the soil corrosion problem is a key and intractable factor that threatens the safe operation of a grounding grid [15,16]. With the continuous exposure of corrosion problems worldwide, a large amount of financial resources and manpower are needed to transform failure grounding devices. For the existing grounding grids, a new corrosion detection and diagnose method is required to understand the corrosion degree of the grounding grid in time and grasp the operating status of the grounding grid. In addition, studying the corrosion of grounding grids and developing new corrosion protection technologies have far-reaching guiding significance for the design and construction of grounding grids in the future. Therefore, further researches on corrosion mechanism, corrosion protection, and new corrosion detection and diagnose methods are required to ensure the rapid development of the power system.

## 2. The Formation Mechanism and Influencing Factors of Soil Corrosion

### 2.1. The Function and Structure of the Grounding Grid

The grounding grid is a facility that realizes the electrical connection between electrical equipment and the ground. The function of the grounding grid mainly includes working grounding, lightning protection grounding, and protective grounding. Working grounding is based on the needs of the normal operation of the power system and is mainly used to stabilize the electrical and circuit ground potential. Lightning protection grounding is a type of grounding that is set to prevent lightning accidents. It protects equipment and people from the hazards caused by lightning discharge when lightning current passes. Protective grounding is mainly used to prevent the accumulation of electric charges on the chassis and the generation of electrostatic discharges that endanger equipment and personal safety. The grounding of the power system can be achieved by the following devices: steel frame of building, single ground rod, multiple ground rods, horizontal ground wire, and grid ground system. Among all the grounding devices, the grid ground system is the most effective one. Therefore, most power plants and substations use grid grounding systems (referred to as a grounding grid) to achieve discharge and voltage equalization. The grounding grid is usually made of copper, steel, and other conductor materials arranged in a grid and laid horizontally. The model of the network ground system is shown in Figure 2. Its area is roughly equivalent to the total area of a power plant or substation. The electrical equipment is connected to the grounding grid through the grounding down conductor to achieve better discharge and voltage equalization.

### 2.2. Material Selection of Grounding Conductor

In order to accommodate the harsh and complex soil environment and working conditions, the selection of grounding conductors must meet the following performance requirements.

Conductivity: The grounding conductors must have good electrical conductivity to meet the function of voltage equalization and current release. The lower the material resistivity, the better the discharge effect of the grounding grid.Corrosion resistance: Given that grounding conductors are placed for a long time under a soil environment with complex physical and chemical properties, the corrosion resistance of the grounding conductor directly affects its work efficiency and service life.Thermal stability: When power equipment is struck by lightning or when a short-circuit fault occurs, the instantaneous large current passing through the grounding grid increases the temperature of the grounding conductor. Therefore, thermal stability verification must be carried out when designing the grounding grid and selecting the grounding conductor.Mechanical properties: Grounding conductors can be divided into two types, i.e., horizontal grounding conductors and vertical grounding conductors. When horizontal grounding conductors are buried in the ground, they are subjected to certain pressure from above. For vertical grounding conductors, they bear a certain impact load when driven into the ground. Therefore, the mechanical strength of the grounding conductor should be considered.

In addition, the selection of the grounding body material should also be based on the comprehensive consideration of the corrosivity of the substation′s soil, the layout of the substation, and the economy of the whole life cycle. According to IEEE Std. 80-2000, “Guide for Safety in AC Substation Grounding”, good electrical conductivity and corrosion resistance make copper as the preferred choice for a grounding conductor. More than 50% of the world, such as Europe and the United States, use copper as the grounding conductor material. Nonetheless, carbon steel and galvanized steel are also widely used in China and in many other countries due to the high cost and limited resources of copper.

#### 2.2.1. Copper

As the first choice for grounding conductors, copper has both good electrical conductivity and excellent corrosion resistance. Copper has excellent electrical conductivity, and its electrical resistivity is $1.75\times 10^{-8}\;\Omega \cdot m$, which is only one-eighth that of steel, meaning that it can ensure good discharge effect of the grounding grid. Copper also has strong corrosion resistance. In soils with high oxygen and water content, the copper surface will corrode uniformly, resulting in corrosion products such as CuO, Cu_2_O, and Cu₂(OH)₂CO₃ [17,18]. The corrosion products of copper have the characteristics of high resistivity and strong adsorption, which can effectively protect the internal substrate from further corrosion. According to corrosion tests, the corrosion rate of copper in the soil is about 1/10 to 1/30 that of carbon steel and roughly 1/3 that of galvanized steel, and the corrosion rate of copper is decreasing year by year [19,20,21]. The melting point of copper is 1083 °C. In the event of a lightning strike or short circuit, the maximum allowable temperature of copper is 450 °C, which shows its excellent thermal stability. However, copper is not the perfect choice for a grounding conductor. As a nonferrous metal, copper is scarce in resources and expensive. The engineering cost of the copper grounding grid is 4–6 times that of the steel grounding grid [22]. In acidic soils containing sulfides, copper has poor corrosion resistance, and its corrosion process gradually accelerates [23]. Moreover, copper is easily connected with other underground steel components, which can form a galvanic corrosion cell, causing accelerated damage to the steel components [24,25]. In addition, the heavy metal ions produced after copper corrosion can harm the environment and pollute soil and groundwater. The heavy metal ions can also accumulate in livestock or humans through the food chain, thereby indirectly or directly endangering human health [26,27,28].

#### 2.2.2. Carbon Steel

Carbon steel is widely used in China as an alternative to copper and is gradually accepted by many other countries because of its low cost. However, carbon steel as a grounding conductor faces many problems in actual use. Due to the unevenness of the soil environment, carbon steel is prone to locally corrode in the soil. Iron corrosion products such as FeOOH, Fe_2_O_3_, and Fe_3_O_4_ are mostly loose structures, which do not protect the matrix but instead promote the corrosion process [29,30,31]. Various environmental factors such as moisture, oxygen content, and salt content have a greater impact on the local corrosion of carbon steel. Under low oxygen conditions, various microbial actions also accelerates the basic corrosion rate of steel. In addition, due to the presence of carbon, carbon steel itself is also prone to form corrosion microcells [32,33]. Carbon steel can become brittle, or even break at the corroded parts, which increases the grounding resistance; such a condition fails to meet the functional requirements of the grounding grid. When the grounding grid is subjected to lightning or short-circuit, the corroded parts are likely to cause ground faults and large-scale grid accidents. In addition, the electrical resistivity of carbon steel is higher than that of copper. Choosing carbon steel as a grounding conductor will increase the grounding resistance of the grounding grid, which makes it difficult to meet the ideal discharge effect. The maximum allowable temperature of carbon steel is 400 °C, which shows that the thermal stability of carbon steel is worse than that of copper.

#### 2.2.3. Galvanized Steel

With the exposure of corrosion problems of carbon steel, galvanized steel is increasingly being used as an alternative grounding conductor material for carbon steel. Compared with carbon steel, galvanized steel has the advantages of simple preparation process, low cost, good conductivity, and better corrosion resistance under normal conditions. The zinc-plated layer on the surface can form a dense oxide film to protect the inner steel substrate. When the zinc-plated layer and the steel substrate form a corrosion cell, because the electrode potential of zinc is more negative, zinc corrodes and dissolves as the anode of the corrosion cell, and steel is protected as a cathode [34,35]. However, through actual tests, it is found that the application effect of galvanized steel as a grounding anticorrosion conductor is not ideal. The anticorrosion effect of galvanized steel is mainly realized by the galvanized layer on the surface. The corrosion rate of the zinc layer is greatly affected by external conditions. In a strongly acidic or alkaline soil environment, the corrosion resistance of galvanized steel is worse than that of copper and carbon steel [36,37]. Due to the skin effect of the zinc layer, the evacuation current flowing through the grounding conductor and the stray current in the soil can eventually cause corrosion of the zinc layer, which greatly accelerates the corrosion rate of the zinc layer [22]. In addition, after corrosion or external force causes the zinc layer to be damaged, the exposed steel matrix forms a galvanic corrosion cell with the zinc layer, further accelerating the corrosion rate of the zinc layer [38,39,40]. When the zinc layer of galvanized steel is corroded, its corrosion rate in the soil is no different from that of carbon steel. Therefore, the service life of galvanized steel depends on the thickness of the galvanized layer on the surface. In many countries, the thickness of the zinc layer of galvanized steel is thin, about 0.05 to 0.08 mm, which greatly limits the protective effect of galvanized steel.

#### 2.2.4. Other Grounding Materials

In addition to the three commonly used ground conductor materials, many new conductor materials are now put in use. Copper-plated steel, copper-clad steel, and hot-dip tin-plated copper steel are used as alternative materials to copper, which have the advantages of low cost, good electrical conductivity, nicer corrosion resistance, and mechanical strength [19,25]. However, due to the limited thickness of the copper protective layer, once the protective layer is damaged, the exposed copper and the steel substrate form a galvanic corrosion cell, which then accelerates the corrosion of the steel substrate [19,41]. Therefore, such alternative materials still face great difficulties in practical applications. Stainless steel was used by the former Soviet Union as a grounding conductor. Due to the chromium element in stainless steel, the electrode potential of steel changes from negative to positive, which gives it a good soil corrosion resistance [42,43]. However, in soils with poor air permeability, stainless steel is extremely prone to pitting corrosion. In addition, ${\mathrm{Cl}}^{-}$ ions, ${\mathrm{SO}}_{4}^{2-}$ ions, and various microbial actions in the soil can also promote the corrosion process of stainless steel [12,44]. It should be noted that the cost of stainless steel is higher than that of carbon steel due to its higher alloy content. Moreover, the addition of alloy increases the resistivity of stainless steel, which in turn affects the discharge effect of the grounding grid. There are also many metal grounding materials such as zinc-clad steel, lead-clad steel, stainless steel-clad steel, and non-metal grounding materials such as artificial graphite, all of which are still in theoretical research due to various issues such as practical performance, preparation process, and production cost [45,46].

### 2.3. Working Environment of Grounding Grid

#### 2.3.1. Soil Environment

The grounding grid is buried in the soil 0.5–0.8 m below the ground for a long period of time. Soil is a heterogeneous multiphase system composed of water, air, and soil particles, and such a system contains a variety of organic and inorganic substances. Soil is a colloidal substance, which has a strong ability to absorb and retain water molecules. This is also the main starting point for the study of soil as a corrosive medium and as an ion conductor [47,48]. The main reason for the corrosion of grounding conductors in soil is the electrochemical inhomogeneity of the material and its surrounding medium, which forms a corrosion cell. Various physical and chemical properties of soil, such as moisture content, oxygen content, soluble salt content, electrical conductivity, and pH value, have a significant impact on the corrosion process of grounding conductor in soil [49,50]. The physical and chemical properties of soil are closely related to the soil texture and local climatic conditions, meaning that it has obvious regional characteristics [51]. The interaction of the abovementioned various factors makes the corrosion law of grounding conductor in soil extremely complicated.

#### 2.3.2. Working Conditions

The grounding grid connects a part of power equipment to the earth through the grounding device to ensure the safety of electrical equipment and the discharge of lightning current. Therefore, there are stray currents and electromagnetic fields in the surrounding environment of the grounding grid. In the event of a lightning strike or power failure, a large instantaneous current pass through the grounding conductor and causes the temperature of the grounding conductor to rise. These factors affect the soil corrosion process of the grounding grid. However, due to the short action time and slight influence of pulsed current and magnetic field on the soil corrosion process, there are few studies about them. In addition, stray current has become a research hotspot due to its significant impact on the soil corrosion process.

The sources of stray current in the grounding grid can be divided into five types according to different generating systems:Metal casings of electrical devices, components of power distribution devices, etc., may become charged due to leakage and accumulation of electrostatic charges. Safety protection devices that are set up to prevent them from harming equipment and personal safety might introduce stray currents.A lightning protection device designed to discharge lightning current to the earth will introduce stray current.The working ground for maintaining the stability of the system, such as the neutral grounding of the transformer, will introduce stray current.In the cathodic protection system, the anode serves as the current output terminal to provide protection current to the protected cathode metal structure. If there are other metal structures near the anode, the anode current will preferentially flow into the metal structure from the channel with the lowest resistance between the metal structure and the anode. This part of the current flowing into the metal structure becomes stray current.Due to the increase in the construction of electrified railways and urban subways, grounding grids inevitably appear near electrified railways and subways. Therefore, the stray current leaking from the walking rails to the ground inevitably affects the safety of nearby grounding grids.

### 2.4. Corrosion Mechanism of Soil Corrosion

Corrosion is the primary problem that threatens the safe operation of grounding grids in soil environments. The corrosion of the grounding grid in soil is mainly electrochemical corrosion. Due to the complex physical and chemical properties of soil, the potential of each part of the grounding material buried in soil is different, thus forming a potential difference, which is the fundamental cause of soil corrosion of the grounding grid. The parts of the ground material with different potentials form a circulating loop through the corrosive medium in the soil to form a corrosion cell. The part with a negative potential serves as an anode to cause a metal dissolution reaction. The part with a positive potential serves as a cathode for cathode reactions such as water or hydrogen reduction and oxygen reduction. In addition, galvanic corrosion of the metal is likely to occur due to the stress disequilibrium and discharge function of the grounding grid.

Anode reaction

The main anode reactions of three common grounding materials (copper, carbon steel, and galvanized steel) are be expounded below [15,18,52,53].

The anode process of soil corrosion of carbon steel is mainly the dissolution process of carbon steel:(1)$$\mathrm{Fe}\to {\mathrm{Fe}}^{2+}+2e$$

In strongly acidic soils, iron ions are mainly dissolved in soil moisture in the form of ${\mathrm{Fe}}^{2+}$ and ${\mathrm{Fe}}^{3+}$ hydrated ions. In stable neutral and alkaline soils, ${\mathrm{Fe}}^{2+}$ continues to react with ${\mathrm{OH}}^{-}$ and $O_{2}$ in the soil to form insoluble hydroxides:(2)$${\mathrm{Fe}}^{2+}+2{\mathrm{OH}}^{-}\to \mathrm{Fe}{\left(\mathrm{OH}\right)}_{2}$$
(3)$$4\mathrm{Fe}{\left(\mathrm{OH}\right)}_{2}+2H_{2}O+O_{2}\to 4\mathrm{Fe}{\left(\mathrm{OH}\right)}_{3}$$

$\mathrm{Fe}{\left(\mathrm{OH}\right)}_{3}$ is unstable and will transform into more stable corrosion products in moist soil:(4)$$\mathrm{Fe}{\left(\mathrm{OH}\right)}_{3}\to \mathrm{FeOOH}+H_{2}O$$
(5)$$\mathrm{Fe}{\left(\mathrm{OH}\right)}_{3}\to {\mathrm{Fe}}_{2}O_{3}+3H_{2}O$$
(6)$$2\mathrm{Fe}{\left(\mathrm{OH}\right)}_{2}+\mathrm{Fe}{\left(\mathrm{OH}\right)}_{3}\to {\mathrm{Fe}}_{3}O_{4}+4H_{2}O$$

Metal cations in the anode zone can also react with anions such as ${\mathrm{HCO}}_{3}^{-},$ ${\mathrm{CO}}_{3}^{2-}$, and $S^{2-}$ in the soil to form insoluble corrosion products:(7)$${\mathrm{Fe}}^{2+}+{\mathrm{CO}}_{3}^{2-}\to {\mathrm{FeCO}}_{3}$$
(8)$${\mathrm{Fe}}^{2+}+S^{2-}\to \mathrm{FeS}$$

For galvanized steel, the anode reaction process is the dissolution of the galvanized layer:(9)$$\mathrm{Zn}\to {\mathrm{Zn}}^{2+}+2e$$

${\mathrm{Zn}}^{2+}$ will continue to react with ${\mathrm{OH}}^{-}$ in the soil to form hydroxides in moist soil:(10)$${\mathrm{Zn}}^{2+}+2{\mathrm{OH}}^{-}\to \mathrm{Zn}{\left(\mathrm{OH}\right)}_{2}\to \mathrm{ZnO}+H_{2}O$$

Copper is a material resistant to soil corrosion and generally undergoes uniform corrosion. The main anodic reactions are as follows:(11)$$\mathrm{Cu}\to {\mathrm{Cu}}^{+}+e$$
(12)$${\mathrm{Cu}}^{+}+{\mathrm{OH}}^{-}\to \mathrm{Cu}\left(\mathrm{OH}\right)\to \frac{1}{2}{\mathrm{Cu}}_{2}O+\frac{1}{2}H_{2}O$$
when ${\mathrm{Cl}}^{-}$ ions are present in the soil:(13)$${\mathrm{Cu}}^{+}+{\mathrm{Cl}}^{-}\to \mathrm{CuCl}$$

It is possible that ${\mathrm{Cu}}_{2}O$ further reacts with $O_{2},$ $H_{2}O$, and ${\mathrm{CO}}_{2}$ in the environment:(14)$$2{\mathrm{Cu}}_{2}O+O_{2}\to 4\mathrm{CuO}$$
(15)$$2\mathrm{CuO}+{\mathrm{CO}}_{2}+H_{2}O\to {\mathrm{Cu}}_{2}{\left(\mathrm{OH}\right)}_{2}{\mathrm{CO}}_{3}$$

2.Cathodic reaction

In strong acid soil, the cathodic process of metal corrosion is mainly a hydrogen evolution reaction:(16)$$2H^{+}+2e\to H_{2}↑$$

In weakly acidic and alkaline soils, the cathodic process is mainly the depolarization process of oxygen:(17)$$O_{2}+2H_{2}O+4e\to 4{\mathrm{OH}}^{-}$$

In soils containing sulfate-reducing bacteria, the cathodic process of soil corrosion may also be sulfate reduction:(18)$${\mathrm{SO}}_{4}^{2-}+2H_{2}O+8e\to H_{2}S+2{\mathrm{OH}}^{-}$$

Metal (M) obtains electrons and transforms from high-valent ions into low-valent ions, which is also a cathodic process of soil corrosion:(19)$$M^{3+}+e\to M^{2+}$$

### 2.5. The Corrosion Type of Soil Corrosion

Ground grid corrosion is greatly accelerated by the uneven electrochemical properties of the grounding material and the surrounding medium. In a complex soil environment, the surface of the grounding material often produces a variety of different corrosion states due to changes in the physical and chemical properties of the surrounding medium, in turn transforming the cathode and anode reaction of the electrochemical corrosion and then causing multiple forms of corrosion. The soil corrosion of the grounding grid mainly includes four types: macro cell corrosion, micro cell corrosion, stray current corrosion, and microbial corrosion.

#### 2.5.1. Macro Cell Corrosion

Macro cell corrosion might be caused by soil inhomogeneity. Macro cell corrosion mainly occurs in a large-span grounding grid structure. The natural corrosion potential of grounding materials at different locations is different due to the heterogeneity of soil porosity, permeability, salt content, temperature, texture, and other properties. As a result, a potential difference is formed between different parts of the grounding material buried in soil in a large area, causing local corrosion.

Taking the formation of an oxygen concentration cell due to the different oxygen permeabilities in the soil as an example, when the grounding material is in soil that has different oxygen concentration content, a macroscopic oxygen concentration cell is formed. The grounding material buried in the compact and moist soil has low corrosion potential due to lack of oxygen, and it acts as an anode to accelerate corrosion. The grounding materials buried in loose, dry soils have high corrosion potentials due to oxygen enrichment and serve as cathodes to slow down corrosion. When the grounding grid is buried, since the lower part of the grounding grid is located in dense undisturbed soil and the upper part is located in loose backfilled soil, the two parts of the grounding grid form a concentration difference cell due to the difference in oxygen concentration. In addition, during the burying of the grounding grid, if there are debris, construction waste, and other debris in the backfilled soil, due to the difference in the permeability of the local medium, oxygen-poor and oxygen-rich areas will also be formed. The part with low oxygen concentration becomes an anode to accelerate corrosion [54]. Due to the high corrosion current density of the oxygen concentration cell, strong local corrosion occurs on the surface of the anode area. The ${\mathrm{Fe}}^{2+}$ generated by corrosion will quickly be converted into $\mathrm{Fe}{\left(\mathrm{OH}\right)}_{3}$ with adhesive effect to prevent the diffusion of oxygen, which further increases the anode activity, eventually forming obvious corrosion pits and even corrosion perforations [55].

In addition, due to the different salt content in the soil, the potential of the grounding conductor at different salt concentration positions is also different. The grounding material at a location with a high salt concentration has a lower potential, and the potential of the grounding material at a location with a low salt concentration is higher, which will constitute a galvanic corrosion cell, and the grounding material with low potential will act as anode to accelerate corrosion [56,57]. If the soil contains organic acids or sulfides, the macro cell corrosion can also occur due to changes of soil properties [12,32,33,58,59].

The connection between different metals will cause galvanic corrosion due to different potentials. Figure 3 shows the galvanic corrosion cell formed by connecting carbon steel and underground copper components. The area ratio of cathode to anode is an important factor in the galvanic corrosion process. Since the corrosion process of most metals in the soil is controlled by the diffusion of oxygen, an increase in the area of the cathode increases the amount of oxygen reduced on its surface. The number of circulating electrons then increases, and the intensity of corrosion current also increases, resulting in faster corrosion [19,41,56]. Galvanic corrosion can also occur when new materials are used to replace rusty materials to repair damaged grounding grids. Due to the different natural corrosion potentials of the new and old materials, the two will form a galvanic corrosion cell when they are connected to each other. The galvanic corrosion cell formed by the contact of new and old grounding materials is shown in Figure 4. Due to the presence of the corrosion product layer on the surface, the rusted material has a higher potential and acts as the cathode of the corrosion cell, and the new material acts as the anode to accelerate corrosion [19].

The grounding material is subjected to various stresses in the grounding grid. For this reason, the grounding material will be deformed to meet the requirements for laying the grounding grid, and the heavily deformed parts will cause stress concentration. In daily operation, due to the surrounding magnetic field and the current in the grounding conductor, the grounding material is subjected to uneven electromagnetic force. In addition, the soil and buildings above also subject the grounding material to uneven pressure. This stress difference then produces a potential difference, which in turn forms a stress corrosion cell [13,60]. The low stress area is protected as the cathode, and the high stress area becomes the anode to accelerate corrosion.

#### 2.5.2. Micro Cell Corrosion

Micro cell corrosion is a kind of corrosion caused by the uneven microelectrochemical state on the surface of the grounding material. When the soil properties are uniform or the size of the grounding material is small, the main form of corrosion is micro cell corrosion. In the case of steel as an example, there are impurities in the steel, such as carbon, and so there are Fe and C on the surface of the steel that are tightly combined. When in contact with soil (equivalent to electrolyte), Fe, C, and the soil form a micro cell [30,49]. Micro cell corrosion proceeds in a microscopic state, and its main form of corrosion is uniform corrosion. The corrosion reaction of the micro cell is weak and does not cause serious harm. Due to the particularity of the soil complex system, it is difficult to achieve a completely uniform medium environment. Normally, the soil corrosion of the grounding material is the result of the joint action of macro cell corrosion and micro cell corrosion.

#### 2.5.3. Stray Current Corrosion

Stray current is a current that is not fixed in size and direction, and it flows into the soil due to discharge from a normal circuit. It can be divided into two types: DC stray current and AC stray current. The corrosion of grounding materials caused by this current is called stray current corrosion. Since the main function of a grounding grid is to release various unbalanced currents in the circuit to the earth, including lightning current, fault current, induced current, etc., the corrosion of grounding materials by stray currents should be the focus of research in this field.

Stray current corrosion is a type of electrochemical corrosion. DC stray current is mainly generated by a DC power transmission system, a cathodic protection system and other power systems with DC power sources [61]. The corrosion mechanism of DC stray current is similar to electrolytic corrosion. The corrosion damage area of DC stray current is relatively concentrated. It can cause rapid corrosion of the ground grid structure, and the corrosion effect on the ground grid material is more severe than other environmental factors. As shown in Figure 5, the part where the stray current flows into the grounding material is protected as the cathode, and the part where the stray current flows out is used as the anode to accelerate corrosion. The DC stray current is analyzed through the schematic of polarization curves (Figure 6). In fact, the stray current polarizes the self-corrosion potential anode from $\varphi_{c}$ to $\varphi_{c}^{\prime }$; the corresponding corrosion current increases from $i_{c}$ to $i_{c}^{\prime }$, and $i_{c}^{\prime }-i_{c}$ is the amount of corrosion increased due to stray current. The amount of metal corrosion caused by DC stray current can be calculated according to Faraday’s law. According to calculations, 1 A current can approximately corrode 9.1 kg iron, 20.7 kg copper, and 4.0 kg magnesium in one year. It can be seen that the DC stray current can cause serious corrosion damage to the grounding grid metal [62,63,64].

AC stray current is generally caused by the secondary induced alternating current generated by the high-voltage power line near the grounding grid superimposed on the grounding grid. Since the AC current passes through the grounding grid for a short time, it is generally believed that the corrosion of AC stray current is less harmful than that of DC. However, the concentrated corrosion of AC stray current is strong. For example, under the action of strong electric fields such as high voltage and ultrahigh voltage transmission grids, the concentrated corrosion effect of AC stray current is particularly serious [64,65].

#### 2.5.4. Microbial Corrosion

The microorganisms in the soil do not directly participate in the corrosion process of metals, but indirectly they cause the corrosion of grounding materials through their life activities. Microorganisms can produce metabolites such as sulfuric acid, organic acids, and sulfides, all of which can change the environmental conditions of grounded metals, such as pH and salt concentration; Some microorganisms such as sulfate-reducing bacteria can directly affect the kinetic process of the electrode reaction; the life activities of some microorganisms may also destroy the protective layer on the metal surface [12,32,33,44].

Sulfate-reducing bacteria are one of the most important factors that cause microbial corrosion. In an anoxic soil environment, sulfate-reducing bacteria multiply in large numbers and promote the cathodic depolarization reaction of the corrosion process through biological catalysis, which greatly accelerates the corrosion of metals [66,67]. Its principle of action is shown in Figure 7. In addition to sulfate-reducing bacteria, thiobacillus is also a microorganism that can affect soil corrosion. Thiobacillus includes two types: thiobacillus thiooxidant and thiobacillus thioparus. Thiobacillus thiooxidant oxidizes the sulfur element produced by thiobacillus thioparus to sulfuric acid to acidify the metal environment and accelerate the corrosion of the metal.

### 2.6. Influencing Factors of Soil Corrosion

#### 2.6.1. Physical and Chemical Properties of Soil

*Soil moisture content.* The moisture content of the soil is a key factor affecting the corrosion of metal materials. Moisture is a prerequisite for soil to become an electrolyte and cause electrochemical corrosion. The research results show that as the soil moisture content increases, the soil corrosiveness increases, and there is a maximum in the promotion of the moisture content on the corrosion of grounding materials [68]. Taking carbon steel as an example, as the moisture content increases, the corrosion rate of Q235 carbon steel in simulated soil first increases and then decreases. When the moisture content reaches 20% to 45%, the corrosion rate is the highest [50,68,69]. The corrosion rate of grounding materials in soils with extremely high and low moisture content is relatively small. The corrosion process of the grounding material in the soil is regarded as a corrosion battery. When the moisture content is low, the increase in the moisture content makes the circuit resistance of the corrosion battery smaller and the soil corrosivity increases. Until a certain critical value, all soluble salts in the soil are dissolved, and the loop resistance reaches the minimum. When further increasing the moisture content, the pores of the soil are filled with water, the oxygen content and soluble salt concentration are reduced, and the soil corrosion is reduced.

*Soil oxygen content.* Oxygen is an important factor that affects metal corrosion. It is directly related to the progress of the cathodic process of soil corrosion. It also indirectly affects the anode process by affecting the difficulty of metal film formation. Because the depolarization of oxygen increases with the increase of oxygen content, the oxygen content of the soil indirectly affects the soil corrosion process by changing the electrochemical cathode and anode reaction rate. At the same time, the oxygen content also affects the metal corrosion electrode potential in the soil to affect the corrosion of the soil. In the initial stage of corrosion, the oxygen content promotes the corrosion of the grounding materials. As the corrosion progresses, copper and carbon steel are more likely to form a protective rust layer in the soil with good air permeability, slowing down the corrosion rate [70,71,72]. The oxygen content in the soil is related to many factors, such as porosity, moisture content, particle size, etc. [71,73]. Sandy soil has a large porosity, and the larger the particle size of the sandy soil, the higher the oxygen content in the soil; loam and clay have fine soil particles, small porosity, and good water retention, so their oxygen content is also low. The oxygen content in the soil is not equal to the dissolved oxygen in the corrosive medium in the soil. The latter is the factor that affects the corrosion of metal materials in the soil. The water content of the soil has a relatively large impact on the dissolved oxygen content. Water content that is too high or too low will reduce the dissolution of oxygen in the medium, inhibit the depolarization reaction of cathodic oxygen, and reduce the rate of soil corrosion.

*Soil resistivity*. For soil corrosion field and power system grounding grid design, soil resistivity is an important consideration. From the perspective of electrical science, the lower the resistivity of the soil, the better the discharge effect of the grounding grid; from the perspective of corrosion science, the lower the resistivity means that the charge transfer is easier and that the corrosion is easier to proceed. Using soil resistivity to evaluate soil corrosion is a very convenient method. Many countries have developed corresponding criteria to evaluate soil corrosion grade by soil resistivity according to their own conditions (see Table 1) [50,74,75]. However, the soil resistivity and the corrosion rate of metals in the soil are not simply a single correlation. For soil corrosion dominated by macro cell corrosion, especially when the cathode and anode are far apart, the resistivity plays a major role, but when the micro cell corrosion plays a dominant role, the soil resistance between the cathode and anode can be ignored since they are in the same position. Therefore, it can be said that it is not a single factor that affects soil corrosion but the combined effect of multiple factors. Resistivity is a comprehensive parameter. Changes in soil type, soil moisture content, and soluble salt content all lead to changes in resistivity. The research of Yoon G. and Fu J. showed that moisture content is the biggest factor affecting resistivity [76,77].

*Soil soluble salt*. Soluble salt is an important part of the soil electrolyte. From an electrochemical point of view, it not only plays the role of transport and conduction, but it also has an impact on soil corrosion by participating in electrochemical reactions. The content of soluble salt in the soil directly affects other physical and chemical properties of the soil. As the content of soluble salt increases, the resistivity of the soil decreases, and the solubility of oxygen in the soil also decreases. Generally speaking, as the soil salt content increases and the resistivity decreases, the macroscopic corrosion rate of grounding materials increases [45]. However, there are many types of soluble salt in soil, and different types of salt have different effects on corrosion. Among them,${\;\mathrm{Cl}}^{-}$ and ${\mathrm{SO}}_{4}^{2-}$ have the greatest impact on metal corrosion. ${\mathrm{Cl}}^{-}$ greatly damages the passivation layer of metal materials. It can penetrate through the corrosion layer of metal and react with the metal matrix to generate corrosion products, which promote the anode process of soil corrosion. Liu et al. found that the severity of stainless-steel corrosion varies with the chloride concentration through corrosion exposure experiments, and the corrosion potential has a linear relationship with the logarithm of the chloride concentration [78]. The higher the chloride concentration in the soil, the more corrosive the soil, which has been confirmed in many studies [11,54,79,80,81]. Karimian et al. found that ${\mathrm{SO}}_{4}^{2-}$ in the soil can be used as a catalyst to participate in the oxidation of iron and promote the corrosion process [82]. In addition, the sulfate-reducing bacteria present in the soil can also cause severe corrosion of metals [23,33,44]. ${\mathrm{CO}}_{3}^{2-}$ has an important influence on the corrosion of carbon steel. ${\mathrm{CO}}_{3}^{2-}$ can generate ${\mathrm{CaCO}}_{3}$ with ${\mathrm{Ca}}^{2+}$ and combine with the sand in the soil to form a strong protective layer, which effectively inhibits the anodic process of corrosion and slows down the corrosion of carbon steel [11,54,80]. The cations such as ${\mathrm{Na}}^{+}$, $K^{+}$, and ${\mathrm{Al}}^{3+}$ in the soil mainly play a conductive role and do not directly affect the electrode process of soil corrosion, so they have little effect on soil corrosion. Cations such as ${\mathrm{Ca}}^{2+}$ and ${\mathrm{Mg}}^{3+}$ can easily form insoluble oxides and carbonates in nonacidic soils, and they can form a protective layer on the metal surface to slow down corrosion.

*Soil pH value*. Soil pH affects the electrode potential of metals by changing the total amount of $H^{+}$ in the soil and their activity [36,37,72]. The main sources of $H^{+}$ in the soil are active $H_{2}{\mathrm{CO}}_{3}$ produced by dissolving ${\mathrm{CO}}_{2}$ in water, organic acids produced by the decomposition of organic matter, and $H_{2}{\mathrm{SO}}_{4}$ produced by oxidation of sulfides. As the soil pH decreases, the $H^{+}$ concentration in the soil increases, and the hydrogen reduction reaction appears and gradually becomes the main cathodic reaction rather than oxygen reduction. The increase of $H^{+}$ concentration can promote the cathodic depolarization process of hydrogen ions, thereby accelerating the corrosion process. As the soil pH increases, the concentration of ${\mathrm{OH}}^{-}$ in the soil increases, and the oxygen reduction becomes the main cathodic reaction.

#### 2.6.2. Stray Current

Because the main function of the grounding grid includes ensuring the leakage safety of power equipment and the discharge of lightning current, there are various stray currents in the soil environment around the grounding grid.

DC stray currents can cause serious corrosion of grounded metals. Studies have shown that the quality loss of metal materials subjected to corrosion by DC stray current conforms to Faraday′s law. The time and intensity of the stray current are the decisive factors affecting its corrosion. In the case of stray current passing through, the corrosion quality loss of grounded metal can be expressed as [22]:(20)$$∆W=\mathrm{kIt}+v_{f}\mathrm{St}$$
where ∆W is the corrosion mass loss of the metal material over a period of time, k is the electrochemical equivalent of the metal material, I is the current flowing into the grounding electrode, t is the corrosion time, $v_{f}$ is the self-corrosion rate of the metal without current, and S is the exposed area of the metal material in the soil. The calculation formula of electrochemical equivalent k is as follows [22]:(21)$$k=\frac{A}{\mathrm{nF}}$$
where A is the atomic weight of the metal, n is the valence state where the metal loses electrons and becomes a metal ion during electrolysis, and F is the Faraday constant.

Studies have shown that the corrosion caused by stray currents, especially DC stray currents, is very serious, and the amount of corrosion is proportional to the current passing time and current intensity [22,60,63,83]. Xu et al. [84] found that DC stray currents have a more serious impact on grounding grid materials than AC stray currents and other environmental factors. They tend to be concentrated in local locations with low resistance and easy discharge, especially those with conductive anticorrosion coatings. The grounding grid can easily form large cathodes and small anodes, which speeds up local corrosion. Wang et al. [85] studied the corrosion behavior of Q235 steel in stray current soil, and they found that stray current would accumulate around Q235 steel, causing Q235 steel to undergo anodic dissolution. With the increase of stray current, both the corrosion current density and corrosion potential increase, thereby aggravating the corrosion rate. Fu et al. [86] studied the corrosion performance of steel by using electrochemical experiments to simulate stray currents, and they found that the application of stray currents causes the corrosion potential of steel to shift negatively, causes the oscillation of anode current density, and reduces the passivation performance of steel.

Compared with metal corrosion caused by DC, the corrosion caused by AC is much milder. Therefore, there are few studies on the corrosion caused by AC stray current. At present, they mainly focus on the preliminary research of AC corrosion mechanism and AC interference parameters (AC voltage, AC current density, AC current frequency).

Mccollum [87] pointed out that the AC corrosion rate decreases as the frequency of the AC current increases, but there is a gate threshold for the frequency. Below this gate threshold, the corrosion rate decreases with the increase of the AC frequency, and above this gate threshold, the corrosion rate increases with the increase of the AC frequency. Marsh [88] found that the influence of AC current density and AC voltage on metal corrosion cannot be ignored, as it affects the passivation performance of metal. Reyes [89] studied the effects of AC and DC voltages on the corrosion system through experiments, and they found that the electric field has a depolarizing effect on both the anode reaction and the cathode reaction, thereby reducing the passivation of the material and accelerating the corrosion rate of the metal. Wen [90] studied the corrosion of coated pipeline steel by alternating current and found that as the stray current density increases, the corrosion potential and corrosion rate increase.

#### 2.6.3. Climatic Conditions

The effects of the above factors on soil corrosion are all affected by changes in climatic conditions. Even with the same type of soil in the same area, its physical and chemical properties are not static. Atmospheric temperature, ventilation, rainfall, evaporation and other climatic conditions will have a certain impact on the soil moisture content, oxygen content, resistivity, microbial activity, etc., and these conditions then affect the soil corrosion process of grounding materials. Therefore, the factors affecting soil corrosion are not constant but have periodic and seasonal changes.

## 3. The Detection Technology and Protection Measures of Soil Corrosion

### 3.1. Detection Technology of Soil Corrosion

Due to the grounding grid being subjected to soil corrosion and to building pressure all year round, grounding conductors of the grounding grid are typically corroded and deformed to varying degrees, and the grounding performance gradually deteriorates. However, only a few reachable nodes can be connected to equipment on the ground through the ground down conductor. Therefore, the corrosion problem of the grounding grid is well hidden from sight, which brings great difficulties to the detection of the corrosion fault of the grounding grid. In general engineering conditions, the grounding grid needs to be tested and maintained when the grounding resistance does not meet proper standards, an accident caused by the grounding grid occurs, or the grounding grid has been in operation for more than 15–20 years.

The research on grounding grid fault diagnosis technology began in the 1980s. At first, it simply measured the grounding impedance of the grounding grid to determine the operating status of the grounding grid. With the continuous research and development of fault diagnosis technology, the current grounding grid fault diagnosis technology is mainly based on the three principles of electromagnetic field, electrical network, and electrochemistry. With the emergence of new technologies, fault diagnosis technologies combining matrix theory, ultrasonic testing, electromagnetic imaging, artificial intelligence, neural networks and other high-tech technologies are also being gradually developed and applied [7,91,92,93,94,95], as we try to obtain as much information as possible about the corrosion status of the grounding grid in the case of uninterrupted power and no excavation.

#### 3.1.1. Electromagnetic Field Method

The fault data are obtained by detecting the ground magnetic field and electric potential difference, and the electromagnetic field analysis is used to obtain the location information of the grounding grid branch breakage, which solves the problem of grounding grid fracture fault location. The measurement accuracy of the electromagnetic field method is limited by the magnetic field sensor. In engineering practice, it is difficult for the sensor to ensure that the moving route during measurement is consistent with the direction of the grounding grid branch. At the same time, the complex electromagnetic fields around power equipment will also interfere with the accuracy of the measurement [96,97]. Yu et al. proposed a transient electromagnetic method to detect grounding grid faults (GG-TEM) [98]. The basic principle diagram of GG-TEM is shown in Figure 8. The pulse current output by the transmitter excites the magnetic field through the coil. When the magnetic field is turned off, the secondary eddy current field is excited at the grounding grid and is confined in the grounding grid to attenuate. The size and attenuation rate of the secondary eddy current field is measured by the receiving coil at a pulse frequency, and these characteristics of the secondary eddy current field are affected by the thickness of the grounding conductor material and its corrosion and electromagnetic properties. Therefore, the corrosion information of the grounding conductor material can be obtained by analyzing the secondary eddy current magnetic field [92,98].

#### 3.1.2. Electrical Network Method

The principle of the electrical network method is to process the grounding network into a pure resistance network and assess the corrosion or cracking of the node through the voltage and ground impedance changes between the different grounding down conductors. Telegen′s theorem and sensitivity matrix are the most important methods in electrical network methods. Because the electrical network method requires the ground lead wire as the observation point of the resistance network, its measurement range and accuracy are limited by the position and number of the ground lead wire. If the number of leads on the grounding grid is small and the locations are concentrated, the number of independent equations in the linear equations formed by the electrical network method will be small, and it will be difficult to solve the resistance value of each branch, which will cause difficulty in fault diagnosis [7,91,99,100]. Hu and Zeng [101,102] put forward a grounding grid fault diagnosis system based on Telegen′s theorem. The equations of port resistance and conductor resistance are established by measuring the resistance between the reachable nodes, and the actual resistance of each conductor of the grounding grid is obtained by an optimizing iterative method. Cao, Li et al. [45,103] assessed the fault of the grounding network by comparing the measured value of the accessible node voltage of the grounding grid with the calculated value of the voltage when there is no fault, and they used a BP neural network to realize an intelligent fault location. Zhang et al. [99] proposed a method of grounding grid fault diagnosis based on a square wave of grounding system frequency model. A real grounding grid model with ten observation points proposed by Zhang et al. is shown in Figure 9. Their research shows that by analyzing and comparing the surface potential of the observation point at different frequencies, it is possible to detect and locate the fault of the grounding grid without excavating.

#### 3.1.3. Electrochemical Method

The principle of the electrochemical method is to measure the electrochemical characteristics of the corrosion reaction between the metal and the soil through the sensor, and then calculate the corrosion rate of the metal and predict the corrosion state of the grounding grid, using approaches including the polarization curve method, potentiometric method, electrochemical impedance spectroscopy, and electrochemical noise [9,104,105]. The electrochemical method has great limitations in the application of grounding grid fault diagnosis [105]. Excavation is needed to place the sensor around the grounding conductor. The location distribution and placement density of the sensor have a great influence on the accuracy of the detection result. In addition, it is difficult to guarantee the measurement accuracy of the sensor due to the interference of the surrounding environment.

### 3.2. Protection Measures of Soil Corrosion

Over time, the surface of the grounding conductor in the soil forms corrosion products due to corrosion, which not only consumes metal materials and reduces the effective cross-sectional area of the conductor, but also forms a physical separation between the grounding conductor and the soil to increase the grounding resistance. Therefore, soil corrosion causes a serious threat to the normal operation of the grounding grid, and it is very necessary to take corresponding protective measures. Since the grounding conductor material requires good electrical conductivity while reducing the corrosion rate, it is imperative to consider both corrosion resistance and electrical conductivity when choosing corrosion protection measures.

#### 3.2.1. Increase the Cross-Sectional Area of the Grounding Conductor

Estimating the amount of corrosion of the grounding conductor during its service life and increasing the cross-sectional area of the grounding conductor to compensate for material corrosion is the simplest and most direct protection method. However, increasing the cross-sectional area not only increases the material cost, but it also increases the difficulty of processing and construction.

#### 3.2.2. Cathodic Protection Technology

Cathodic protection is one of the most commonly used electrochemical protection techniques, including the impressed current method and the sacrificial anode method [106,107]. The impressed current method relies on an external power source to provide cathodic protection current to the grounding grid. The output current of this method can be adjusted manually, and the protection effect is good, but the maintenance workload is heavy. The sacrificial anode method is to provide current for the protected metal through the dissolution of the metal with a more negative potential. The sacrificial anode structure is easy to install and has the features of convenient operation and maintenance. However, its protection current range is small and uncontrollable, and often cannot meet the protection current requirements. In addition, the protective effect of cathodic protection measures is greatly affected by the soil resistivity. Therefore, the application of the cathodic protection method on the grounding grid is still in the experimental stage, and the actual application remains few in number [108,109].

#### 3.2.3. Conductive Anticorrosion Coating

The ideal conductive anticorrosive coating can have both conductivity and corrosion resistance, and it can have a good corrosion protection effect on the basis of meeting the performance requirements of the grounding grid. On the basis of maintaining the corrosion resistance of the traditional anticorrosion coating, the resistivity of the coating is reduced by adding conductive particles. The conductive anticorrosion coating of the grounding grid is classified into four types according to the conductive additives: nickel powder type, graphite type, nanocarbon type, and organic conductive polymer type. The higher the content of conductive particle filler, the better the conductivity of the coating, but the poorer the film-forming ability and compactness of the coating, which in turn leads to a decrease in corrosion protection performance. Therefore, on the basis of ensuring conductivity, reducing particle size and improving particle dispersion to improve corrosion resistance has become the key point of conductive anticorrosion coating [110,111,112].

At present, the research focus of new conductive coatings is focused on carbon nanotubes. Adding carbon nanotubes to the coating while maintaining good dispersion of the carbon nanotubes can greatly increase the conductivity of the coating without reducing the corrosion resistance of the coating. Takahiro et al. [113] prepared a single-walled carbon nanotube coating, and the better the dispersion performance, the better the conductivity of the coating. Gao et al. [114] used single-walled carbon nanotubes and polymer conductive composites to prepare high-conductivity coatings. Huang et al. [115] used spraying technology to prepare the carbon nanotube conductive coating so that the carbon nanotubes formed a staggered conductive structure on the substrate.

#### 3.2.4. New Corrosion-Resistant Grounding Material

Traditional grounding materials include copper, carbon steel, and galvanized steel. Copper is facing challenges of high cost and resource shortage, while carbon steel and galvanized steel are facing serious corrosion problems. Therefore, the use of new corrosion-resistant grounding materials as an alternative to traditional materials can alleviate corrosion problems to a certain extent.

Liu et al. [116] used an adhesive to press graphite and carbon fiber into a shape, and they used a stranding process to prepare a flexible graphite composite grounding material. The material has stable physical and chemical properties and also good corrosion resistance and conductivity.

Adding specific alloying elements to the grounding material can also improve the corrosion resistance of the material. Nishikata et al. [117] studied the corrosion behavior of steel under cyclic dry and wet conditions by using the AC impedance method; they found that the self-corrosion potential of steel increases with the increase of Ni content, and the degree of amorphization and the density of the rust layer also improve. Qian et al. [118] used periodic dry-wet cycle corrosion tests to study the effects of different contents of Cr on steel; they proved that the presence of Cr improves the corrosion resistance of steel, and they found that Cr acts as a barrier to the corrosion process and promotes the formation of protective rust layer. Feng et al. [119] used Fourier transform infrared reflectance spectroscopy, fluorescence spectroscopy, and electrochemical analysis to study the corrosion effect of Mo on carbon steel, and they found that in the soil solution, Mo can accumulate on the pitting active sites of the metal and can weaken the corrosive activity of ${\mathrm{Cl}}^{-}$.

## 4. Conclusions and Prospects

The soil corrosion problem is the result of synergistic action of the soil environment and the working condition of the grounding grid. The corrosion process of the grounding grid accelerates greatly by the inhomogeneity of the soil environment and the existence of stray current. Stray current corrosion is a more serious threat to the ground grid system than other types of corrosion. Corrosion detection and diagnosis technology, such as the electromagnetic field method and the electric network method, is used to detect the corrosion problem of grounding grid in service. Increasing the cross-sectional area of the grounding body and employing cathodic protection technology are currently the most commonly used protection methods. However, the research on soil corrosion of grounding grids still faces many challenges in terms of mechanism, diagnosis, and protection. The following are the prospects for the soil corrosion problem of grounding grids:At present, most studies on soil corrosion of grounding grids focus on soil factors (such as moisture content, oxygen content, salt content, etc.) and ignore the influence of grounding grid working conditions (stray current, high current, impulse current, etc.) on the corrosion process. Combining working conditions with soil factors can reflect the actual corrosion situation of the grounding grid.In the field of grounding grid fault diagnosis, sensors play an important role in sensing current, electromagnetic field, impedance, and transmitting information. Therefore, the development of new sensors or new ways of transmitting information that can resist interference and accurately transmit information is of great significance for improving the accuracy of grounding grid fault diagnosis.At present, the main protection measures against grounding grid corrosion in engineering are to increase the cross-sectional area of the grounding conductor or use galvanized steel as a substitute for carbon steel. These methods have limited protection effects and do not completely solve the corrosion problem. The cathodic protection method also limits its application due to its limited protection effect or excessive maintenance cost. Therefore, the development of an efficient and practical protection method (such as new anticorrosion materials, conductive anticorrosion coatings) is a possible future research hotspot.

## Figures and Tables

**Figure 1 micromachines-12-00513-f001:**
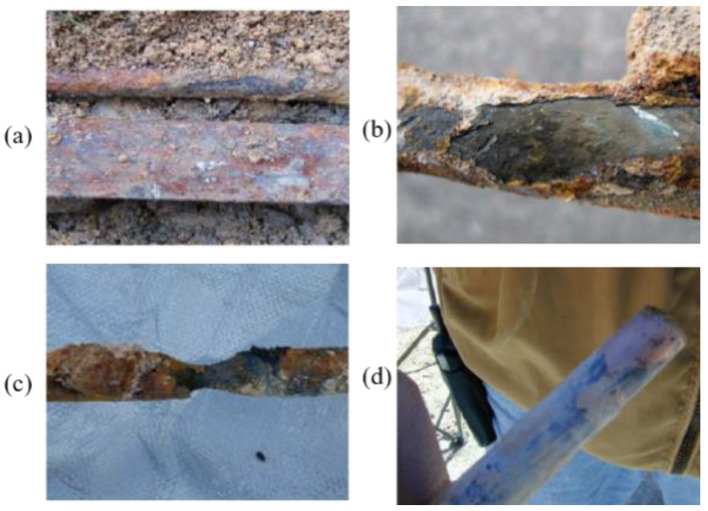
Corrosion of the grounding conductor: (**a**): excavation situation; (**b**): carbon steel; (**c**): galvanized steel; (**d**): copper-clad steel.

**Figure 2 micromachines-12-00513-f002:**
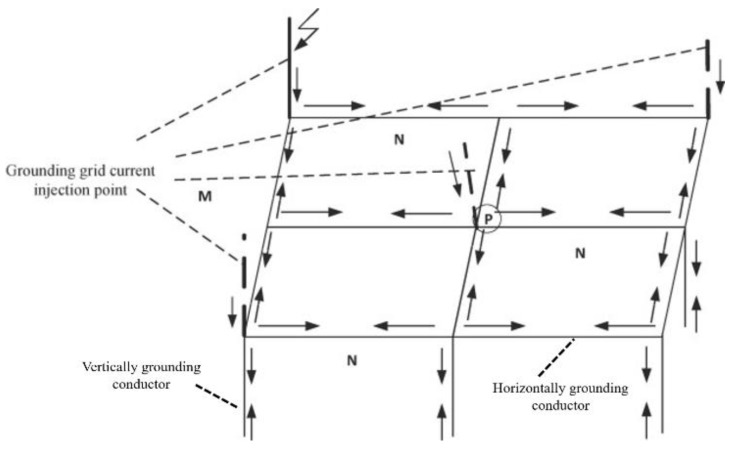
Network grounding system model [2].

**Figure 3 micromachines-12-00513-f003:**
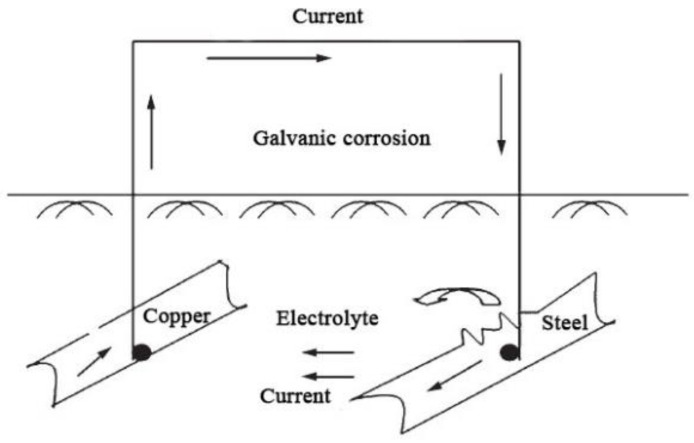
Galvanic corrosion cell formed by the electric contact of two foreign metals.

**Figure 4 micromachines-12-00513-f004:**
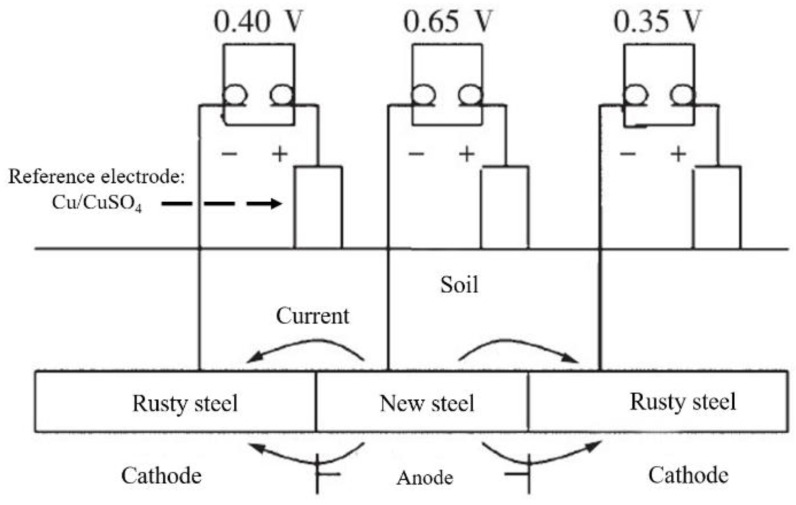
Galvanic corrosion cell formed by contact of new and rusty grounding steels.

**Figure 5 micromachines-12-00513-f005:**
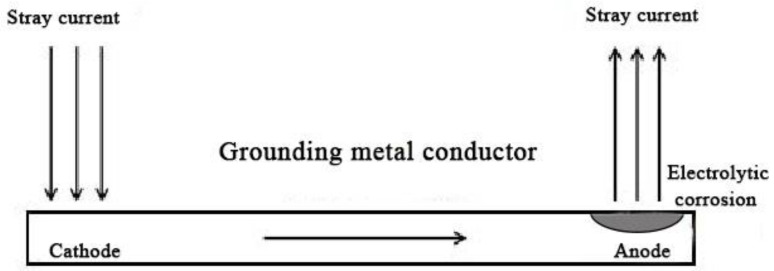
DC stray current corrosion mechanism.

**Figure 6 micromachines-12-00513-f006:**
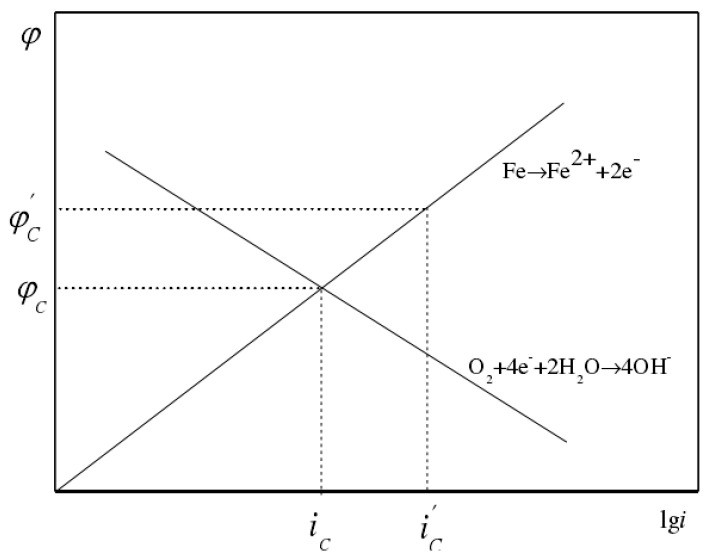
Schematic diagram of the polarization curve of steel with DC stray current.

**Figure 7 micromachines-12-00513-f007:**
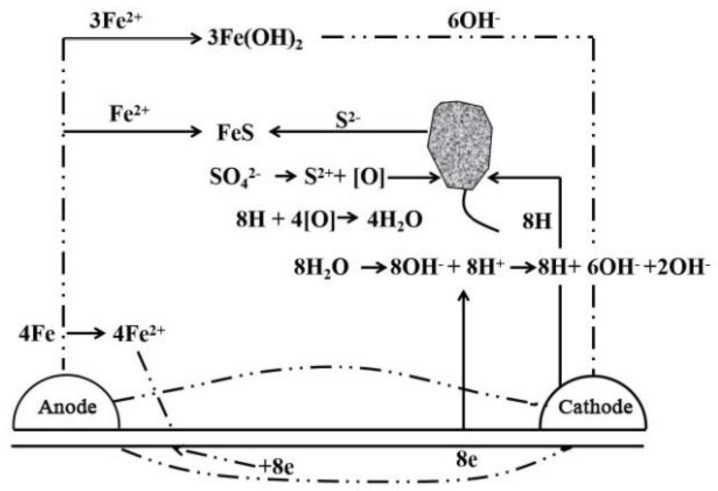
Sulfate reducing bacteria corrosion diagram.

**Figure 8 micromachines-12-00513-f008:**
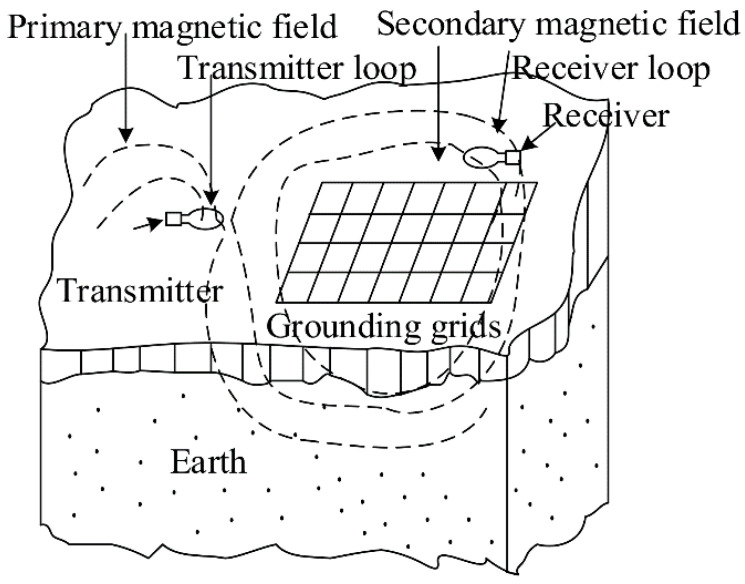
Schematic field setup of the GG-TEM method [92].

**Figure 9 micromachines-12-00513-f009:**
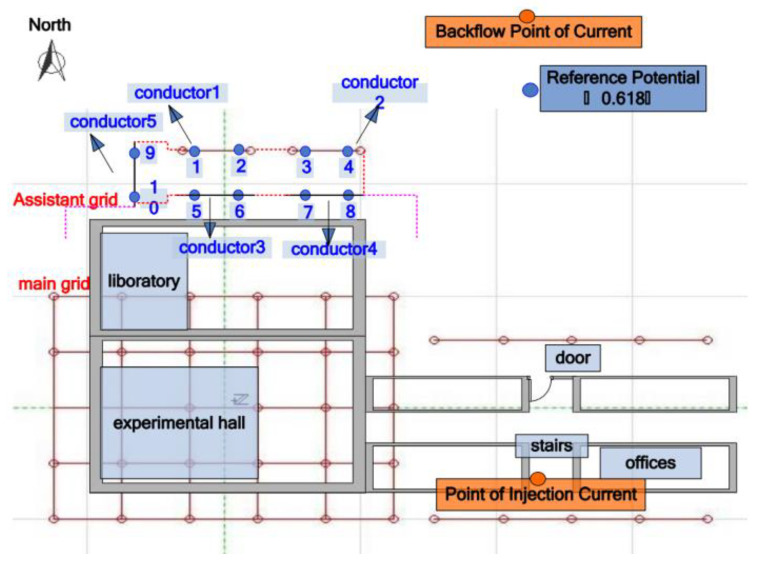
A ten observation points model of an actual grounding grid [99].

**Table 1 micromachines-12-00513-t001:** Standards for evaluating soil corrosion degree based on soil resistivity in different countries.

Corrosion Grade	Soil Resistivity, Ω·m				
	China	US	Japan	France	UK
Extra low	>50	>50	>60	>30	>100
Low			45–60		50–100
Medium	20–50	20–45	20–45	15–25	23–50
High	<20	7–20	<20	5–15	9–23
Extra high		<7.5		<5	<9

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
