# Peer review of "Research Advances of Soil Corrosion of Grounding Grids"

_micromachines, 2021, doi:10.3390/mi12050513_

Round 1

Reviewer 1 Report

In this paper, the causes of soil corrosion problems on grounding grid,  and the influencing factors of corrosion were analysed in detail.It gives only a glance of the soil corrosion problems. Regarding new  corrosion detection and diagnosis technology and better protective measures deeper research is needed.

As a whole Average score shall be given to this paper.

Reviewer 2 Report

The paper is a review paper that intended to summarize the recent research of soil corrosion of grounding grids.  The paper's topic is important and up to date since grounding conductor fault occurring in the power system leads to rising ground potential endanger the secondary equipment and causes electrical shock. 
In general, the paper is well organized and structured. It is easy to read. However, the figures are very informative and helpful to understand corrosion's main mechanisms, but considering the paper is a review paper, a higher number of figures would be preferable to make it easy to understand the details. Please improve the paper with more pictures, especially in section 3.
The paper contains the most important references providing a good reference for curious readers. 
Unfortunately, the paper has a very critical missing point. The authors intended to write a review paper, but they do not provide directions for future research. 
Please improve the paper in this respect, adding a discussion-like section.
All in all, after the revision, this paper could be a good reference work.

Reviewer 3 Report

Comments:

  • Fig. 1 : English needs a revision and the quality of fig. 1 should be improved.
  • - Page 5, line 215: Ref 46 and mainly 47 should be newer since they are related to new materials in use or to future use as grounding materials. 
  • Page 8, line 338: In addition, due to the different salt content in the soil, a salt concentration cell will 338
    be formed, and the grounding material at a location with a high salt concentration has a 339
    lower potential, which accelerates corrosion as the anode of the corrosion cell. Please, explain better and cite references.
  • Page 8, line 340: Please, cite references of thermoelectric cells in soil. Is there a temperature difference that generates thermogalvanic cell?
  • Page 8, line  342: If the soil contains organic acids or sulfides, the macro cell
    corrosion can also occur due to changes in soil properties. Please, give references. 
  • Page 10, line399: Pourbaix ( P should be a capita letter). 
  • Suggestion: Fig. 2 and 4 should show a real situation and Not a drawn scheme.
  • Please, correct: S2- and NOT S2+
  • Table 1: Is the first column Corrosion degree? Please, clarify. 
  • Page 15, line 641: Please, cite references. The authors should say that the cathodic protection method depends on the resistivity of soil.  

Reviewer 4 Report

Review to the article: ” Review Research advances of soil corrosion of grounding grid

The article reviews general knowledge about underground corrosion of different metallic constructions, the kinds of corrosion, and the methods of corrosion protection.

I think the serious mistake of the article is the very general character of publication as in schoolbooks trying to explain corrosion, electrochemical corrosion, and protection. This information is easy to find in corresponding sources. The data relating to grounding grid corrosion and protection are very rare. How to corrode these constrictions (photographs, number of cases). What is the main effective methods of protection of the grids used in practice (e.g. cathodic protection, sacrificial anodes)? You have to be more close to real constructions, give recommendations.

In fact, I do not understand why this article is needed. The impact of new knowledge is very low.

Mistakes and comments

“electrolytic corrosion” . Please remove “electrolytic” in all sentences, it is not used may be galvanic corrosion or just corrosion.

264-268 “The corrosion of the grounding grid in the soil is mainly chemical corrosion and electrochemical corrosion. Chemical corrosion has little effect on the grounding grid, and electrochemical corrosion is the main form of grounding grid corrosion.”

In natural conditions, the corrosion of all metals is electrochemical. Chemical corrosion is high-temperature oxidation by air (oxygen) without water or corrosion of highly uniform alloys like mercury amalgams.

273 “The part with a positive potential serves as a cathode for cathode reactions such as hydrogen evolution or oxygen absorption.”

Cathodic reactions are water or hydrogen reduction and oxygen reduction.

274 “In addition, electrolytic corrosion of the metal will occur due to the pressure equalization and discharge function of the grounding grid.”

The phrase is not clear. What it is pressure equalization?

305 “Ground grid corrosion is mainly the result of the uneven electrochemical properties

of the grounding material and the surrounding medium.”

It is wrong. Corrosion is the result of thermodynamic non-stability of the metal (e.g. steel) and possible reduction of oxidizer (water or oxygen).

Different soil aeration and non-uniformity of alloy composition create galvanic couples. It can only accelerate the corrosion.

338 “In addition, due to the different salt content in the soil, a salt concentration cell will

be formed, and the grounding material at a location with a high salt concentration has a

lower potential, which accelerates corrosion as the anode of the corrosion cell.”

In soils, it can not be found due to quick water and salts penetration to the surface and diffusion of ions equalizing concentration. This kind of couple is not discussed in the literature. It can be possible (local artificial couple) in the air with deposited NaCl crystal or droplet.

349 “amount of oxygen adsorbed on its surface. And then the number of”

Probably replace adsorbed to “reduced”. Adsorption it is just the landing of the molecule on the surface, and reduction is the transformation of O2 molecule to water-hydroxide.

Figure 1, in caption, clarify what it is “observation points”

Figure 2 in the caption add “the electric contact of two foreign metals”.

Figure 3 Add to the caption: voltammeter, reference electrode (??) potential measurements, the parts have to be disconnected to measure current and the potential.

I am not sure that rusted metal will be anode and it will corrode in contact with new material.

It is because rusted steel will be covered by the corrosion products, the potential will increase due to inhibition of anodic reaction of metal oxidation and oxygen reduction will be more effective on the rusted surface. Rusted is the cathode, a new grid is the anode. Please check out and add the reference.

Figure 5 It is not the Paurbaux diagram. It is a schematic of polarization curves anodic and cathodic curves corrosion current and corrosion potential.

394 The sources of DC stray current? Please say on the sources of DC stray current

493” As the content of soluble salt increases, the resistivity of the soil decreases, and the

solubility of oxygen in the soil also decreases. Generally speaking, as the soil salt content

increases and the resistivity decreases, the macroscopic corrosion rate of grounding ma-

terials will increase.”

Please give the references.

An increase in salt content and a decrease in resistivity will transform galvanic corrosion to more uniform corrosion. An increase in the salt concentration normally increases the corrosion rate.

521 “Reaction gradually replaces the oxygen depolarization reaction as the main cathodic reac-

tion. The increase of”

The hydrogen reduction will not replace oxygen reduction. They will go together.

523 “OHin alkaline soil mainly comes from the hydrolysis of CO32−.” It is wrong. Carbonate ion is stable to hydrolyze.

Table 1 is very strange and it is wrong. The resistivity and corrosivity of soil depend not on general geography and country but from a specific place. For example in China, the island in the ocean will have much less resistivity due to the high content NaCl. The place in China in the mountains will have a corrosion rate very low. Remove the table. Probably you mean the standards of the soil resistivity and corrosivity in different countries. Please explain and give the references.

Equations 20 and 21 give the references.

2.12.0.0 2.12.0.0

Round 2

Reviewer 2 Report

Thank you for considering my suggestions. I have no further comments and issues on the paper.

Author Response

Thank you very much for your valuable suggestions. It is a great honor to be recognized by you. 

Reviewer 4 Report

Dear authors, thank you for your work. Just a small comment concerning Figure 4. You have to mark on the schematic: what are the positive squares. Probably these are reference electrodes e.g. Cu/CuSO4. 

In addition the bottom schematic overlay on upper one. Please improve the Figure 4 and publish.  Good luck.
